# Enhancing Adversarial Robustness of Vision Language Models via Adversarial Mixture Prompt Tuning

## Abstract

Large pre-trained Vision Language Models (VLMs) demonstrate excellent generalization capabilities but remain highly susceptible to adversarial examples, posing potential security risks. To improve the robustness of VLMs against adversarial examples, adversarial prompt tuning methods are proposed to align the text feature with the adversarial image feature without changing model parameters. However, when facing various adversarial attacks, a single learnable text prompt has insufficient generalization to align well with all adversarial image features, which ultimately results in overfitting. To address the above challenge, in this paper, we empirically find that increasing the number of learned prompts yields greater robustness improvements than simply extending the length of a single prompt. Building on this observation, we propose an adversarial tuning method named **Adversarial Mixture Prompt Tuning (AMPT)** to enhance the generalization against various adversarial attacks for VLMs. AMPT aims to learn mixture text prompts to obtain more robust text features. To further enhance the adaptability, we propose a conditional weight router based on the input adversarial image to predict the mixture weights of multiple learned prompts, which helps obtain sample-specific mixture text features aligning with different adversarial image features. Extensive experiments across 11 datasets under different settings show that our method can achieve better adversarial robustness than state-of-the-art approaches.

## 1 Introduction

Large pre-trained Vision Language Models (VLMs) such as CLIP Radford et al. (2021) have excellent generalization capabilities and can be regarded as foundation models Bommasani et al. (2021) in different downstream tasks, e.g., image-text retrieval, zero-shot image classification, or image generation guidance. Due to its wide range of application scenarios, it places high requirements on security performance. However, despite its excellent performance, VLMs face many potential security risks Inkawhich et al. (2023); Mao et al. (2022); Schlarmann & Hein (2023), including the fact that visual models are vulnerable to adversarial examples Szegedy et al. (2013), which can pose a serious threat to the application in actual scenarios.

To eliminate this potential security risk, many works have been proposed to improve the robustness of VLMs to adversarial examples, which can be mainly divided into two types, full-parameter fine-tuning Mao et al. (2022); Wang et al. (2024); Schlarmann et al. (2024); Yu et al. (2024) and parameter-efficient fine-tuning Mao et al. (2022); Wang et al. (2024); Zhang et al. (2024); Li et al. (2024); Zhou et al. (2024); Luo et al. (2024). Among them, full-parameter fine-tuning is an effective method to improve the adversarial robustness of the model. However, this method often requires a lot of computational overhead and also affects the performance of the model on general tasks. Another type of parameter-efficient method, e.g., adversarial prompt tuning Zhang et al. (2024), freezes all or most of the weights of the model and only fine-tunes some of its parameters. This type of method can also improve the adversarial robustness with lower training overhead compared with full-parameter fine-tuning, which is a promising solution. However, adversarial prompt tuning faces a serious problem: **insufficient generalization**. For example, for the text prompt tuning Zhang et al. (2024); Li et al. (2024), when only one learnable prompt is fine-tuned, the text feature is not sufficient

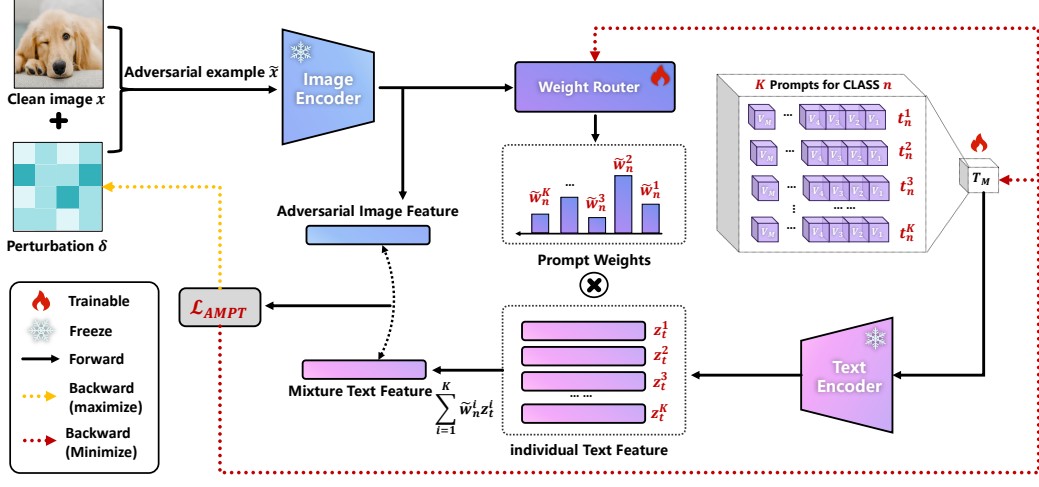

Figure 1: The framework of **Adversarial Mixture Prompt Tuning (AMPT)**. To enhance the adversarial robustness, we apply adversarial mixture prompt to generate diverse individual text feature, and utilize the conditional prompt weight router to obtain a sample-specific mixture text feature, and finally bring more generalization towards different adversarial examples.

to fit the image features for various adversarial examples, which can easily lead to overfitting and further cause potential security risks Wang et al. (2025).

To enhance the generalization of the adversarial text prompt, an intuitive approach is to increase the length of the text prompt. However, we find that when it grows to a certain length, a longer prompt will bring greater optimization difficulty, and also needs higher requirements on the corresponding text encoder to deal with long prompts, finally leading to suboptimal robustness. Inspired by the Mixture of Experts (MoE) paradigm Cai et al. (2024), we adopt an alternative strategy: increasing the number of learnable base prompts. Similar to multiple experts in MoE, we construct a composite prompt by combining several base prompts, with their weights adaptively determined based on the characteristics of the adversarial image. This approach enables the generation of more diverse and expressive text features. Moreover, since each base prompt remains short and is easier to optimize, leading to improved adversarial robustness. A preliminary experiment has empirically verified our idea that increasing the number of prompts can indeed enhance robustness more effectively than simply extending the length of the prompts.

Based on the above consideration, in this paper, we propose an adversarial prompt tuning method named **Adversarial Mixture Prompt Tuning (AMPT)** to enhance the adversarial robustness of VLMs. Specifically, we fix the parameters of the text and image encoders but only optimize adversarial mixture prompts. These text prompts pass through the text encoder and generate diverse individual text features. In addition, to enhance the adaptability, we propose a conditional text weight router based on image features to predict the weights of adversarial mixture prompts and aggregate them into a sample-specific mixture text feature, so as to adaptively align with the diverse adversarial image features. A series of experiments show that our AMPT can achieve better accuracy and robustness than state-of-the-art methods on multiple different datasets. Meanwhile, AMPT also shows better generalization across different datasets. Our contribution can be summarized as follows:

- We find that for adversarial text prompt tuning, increasing the number of learnable text prompts can achieve a better robustness than only increasing the length of learnable text prompts within a certain range of parameters.

- We propose a novel method named Adversarial Mixture Prompt Tuning (AMPT), which applies adversarial mixture prompts to generate diverse individual text features, where each text feature can play its unique roles for different adversarial examples, thereby alleviating the overfitting phenomenon.

- We apply a conditional text weight router based on image features to predict the weights of different text features and obtain a sample-specific mixture text feature that has pretty adaptability to align with different adversarial image features. Furthermore, we theoretically verify the effectiveness of the weight router.

- We empirically verify the effectiveness of AMPT. Extensive experiments demonstrate that our AMPT can outperform state-of-the-art methods against adversarial examples in adversarial robustness and generalization across different datasets.

## 2 RELATED WORK

### 2.1 PROMPT TUNING FOR ACCURACY IN VLMs

Different from the methods of fine-tuning all model parameters, the prompt tuning method only fine-tunes the model's input prompts. Through a training process, a learnable prompt suitable for downstream tasks is obtained to replace the hand-crafted prompt, thereby improving the performance of the VLMs. The prompt tuning methods are originated from text model Li & Liang (2021); Liu et al. (2021) and also have corresponding applications in visual models Jia et al. (2022) and vision-language models Khattak et al. (2023); Zhou et al. (2022b;a). CoOp Zhou et al. (2022b) first utilizes a learnable vector to replace the hand-crafted in Vision-Language Models. Based on CoOp, CoCoOp Zhou et al. (2022a) is proposed by introducing a conditional Meta-net based on an image feature to generate an instance-adaptive vector and add it to the learnable vector. Some research also tries to apply multiple prompts in VLMs Lu et al. (2022); Chen et al. (2022). Different from the above works, in this paper, we mainly focus on improving the adversarial robustness via optimizing multiple prompts and embedding it into the adversarial training framework, which has obvious differences.

### 2.2 ADVERSARIAL PROMPT TUNING IN VLMs

Due to its excellent performance and low training cost, prompt tuning has been applied to improve the robustness of VLMs. Chen et al. (2023) applies visual prompting to enhance the adversarial robustness. Furthermore, TeCoA Mao et al. (2022) and PMG-AFT Wang et al. (2024) employ visual prompt tuning to improve the adversarial robustness of VLMs. AdvPT Zhang et al. (2024) and APT Li et al. (2024) are proposed to apply text prompt tuning to further enhance the VLMs against image attacks. FAP Zhou et al. (2024) tries to enhance the robustness via bimodal tuning, while APD Luo et al. (2024) further extends FAP into the adversarial distillation setting. To solve the insufficient generalization, Wang et al. (2025) applies Test-Time Adversarial Prompt Tuning (TAPT) to learns defensive bimodal (textual and visual) prompts. Different from the above research, this paper improves adversarial robustness through adversarial mixture prompt tuning during training.

## 3 THE NECESSITY OF MIXTURE PROMPTS

### 3.1 FORMULATION OF ADVERSARIAL PROMPT TUNING

CoOp Zhou et al. (2022b) first applies the text prompt tuning in CLIP to improve the performance of downstream tasks, and Zhang et al. (2024); Li et al. (2024) apply the adversarial prompt tuning in improving adversarial robustness, and the optimization goal of adversarial prompt tuning can be defined as follows:

$$\arg\min_t \mathbb{E}_{(x,t,y)\sim\mathcal{D}}(\mathcal{L}(\tilde{x}, t, y; F_{\theta_v}, F_{\theta_t})), \tag{1}$$

where $x$ and $t$ are the image and text pairs belong to the dataset $\mathcal{D}$. For the image classification task with $N$ classes, texts $t$ also contain $N$ different prompts: $\{t_1, t_2, \cdots, t_N\}$. $\tilde{x}$ denotes adversarial examples. $y$ denotes the ground truth. $y_{in}$ indicates whether the image $x_i$ and text $t_n$ pair match, if the image $x_i$ and text $t_n$ match, $y_{in}$ is equal to 1, otherwise $y_{in}$ is equal to 0; the $F_{\theta_v}$ and $F_{\theta_t}$ are the image encoder and text encoder of CLIP.

Meanwhile, as for the text $t$, a fixed text template, e.g., "a photo of a [CLASS]", is often directly used as the text input, and the maximum similarity between it and the input image is calculated to

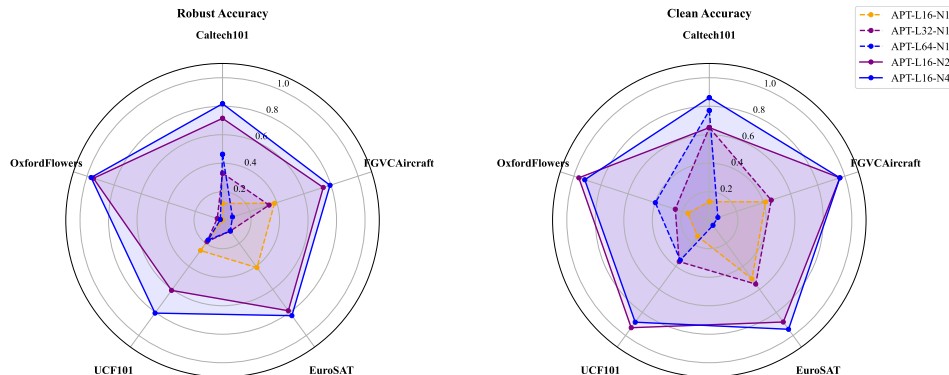

Figure 2: The performance of adversarial prompt tuning with different length and number on five datasets. "APT-L*m*-N*k*" denotes the APT with prompt length **m** and prompt number **k**. We find that increasing the number of prompts can enhance more robustness than increasing the prompt length (i.e., solid lines show better performance than dotted lines).

determine which class the image belongs to. Zhang et al. (2024); Li et al. (2024) apply a learnable text prompt, which consists of the class context and a learnable context and can be defined as follows:

$$t_n = [context_{front}][CLASS_n][context_{end}]. \tag{2}$$

The image feature $z_v^i$ is generated by image encoder $F_{\theta_v}$ of input $\tilde{x}_i$, the text feature $z_t^n$ is generated by text encoder $F_{\theta_t}$ of input $t_n$, which can be defined as follows:

$$\tilde{z}_v^i = F_{\theta_v}(\tilde{x}_i), z_t^n = F_{\theta_t}(t_n). \tag{3}$$

For the image classification task, Cross-Entropy loss is applied as the optimization function in APT Li et al. (2024), which can be defined as follows:

$$\mathcal{L}(\tilde{x}_i, t, y_i; F_{\theta_v}, F_{\theta_t}) = -\sum_{n=1}^{N} y_{in} log \frac{exp(cos(\tilde{z}_v^i, z_t^n))}{\sum_{m=1}^{N} exp(cos(\tilde{z}_v^i, z_t^m))}, \tag{4}$$

where the cosine similarity is applied to measure the degree of alignment of the features, and applying the softmax operation can obtain the probability that the $\tilde{x}_i$ aligns with the $z_t$.

## 3.2 A LONGER PROMPT OR MORE PROMPTS IN APT?

APT Li et al. (2024) extends the CoOp framework to enhance the robustness of VLMs against adversarial attacks. However, a single text prompt has potential generalization problems: when faced with complex adversarial examples, its parameters may struggle to adapt to the change. Therefore, we attempt to explore how to enhance the generalization of adversarial text prompt tuning from the perspectives of increasing length and increasing number.

To compare those two approaches, we keep the total prompt parameters the same. For example, we use a learnable prompt of length 64 to compare the robustness of 4 learnable prompts of length 16. We conducted experiments on five datasets and the results can be viewed in Figure 2. And the experiments are based on APT Li et al. (2024).

The experimental results surprisingly reveal that, during adversarial prompt tuning, increasing the number of prompts is more effective than increasing their length. Specifically, we find that using 2/4 learnable adversarial prompts of length 16 achieves better adversarial robustness compared to using a single prompt of length 32/64, with an average improvement of 3.88%/4.34%. Notably, this setup also leads to an average performance gain of 4.56%/6.43% on clean samples. Furthermore, increasing the number of prompts further enhances adversarial robustness; When the number of prompts increases from 2 to 4, adversarial robustness improves by an additional 0.34%. In contrast, merely increasing the prompt length yields no obvious improvement in adversarial robustness.

We argue that after the total number of parameters reaches a certain level, continuing to increase the length of prompts will increase the difficulty of learning an ideal prompt, so it hardly enhances the adversarial robustness. On the contrary, shorter prompts are relatively easier to learn, and adversarial mixture prompts can generate more diverse text features, which have more possibility to align with adversarial examples. Therefore, increasing the number of prompts can further improve robustness compared with increasing prompt length.

# 4 ADVERSARIAL MIXTURE PROMPT TUNING

## 4.1 OVERALL FRAMEWORK

Based on the above findings, we argue that the generalization of adversarial robustness can be improved by adding adversarial mixture prompts. Therefore, we propose Adversarial Mixture Prompt Tuning (AMPT) to further improve the adversarial robustness of the VLMs. Here our optimization goal can be formulated as follows:

$$\arg\min_{T_m, \theta_w} \mathbb{E}_{(x,t,y) \sim \mathcal{D}}(\mathcal{L}_{AMPT}(\tilde{x}, T_m, y; F_{\theta_v}, F_{\theta_t}, F_{\theta_w})), \tag{5}$$

where $\mathcal{L}_{AMPT}$ denotes the optimization loss function of our AMPT, and $T_m$ denotes the adversarial mixture prompts, $F_{\theta_w}$ denotes the conditional prompt weight router. As for the adversarial examples $\tilde{x}$, we follow the "on-the-fly" setting in Li et al. (2024), where the attacker can access all the parameters of the VLMs including the adversarial mixture prompts but can only apply adversarial perturbations to the image $x$. And the adversarial examples $\tilde{x}$ can be formulated as follows:

$$\tilde{x} = \underset{||\tilde{x}-x|| \leq \epsilon}{argmax} \mathcal{L}_{AMPT}(\tilde{x}, T_m, y; F_{\theta_v}, F_{\theta_t}, F_{\theta_w})), \tag{6}$$

where $\epsilon$ denotes the maximum perturbation scale. It should be mentioned that the "on-the-fly" setting is closer to the adversarial examples in Szegedy et al. (2013); Madry et al. (2017). For the evaluation against adversarial attacks, we also follow this type of setting as default.

## 4.2 ADVERSARIAL MIXTURE PROMPTS

Assume adversarial mixture prompts $T_m$ have $K$ total of prompts, which can be defined as follows:

$$T_m = \{t^1, t^2, \cdots, t^K\}, \tag{7}$$

where $t^k$ denotes the $k$-th learnable adversarial text prompt, which includes $N$ class text prompt: $\{t_1^k, t_2^k, \cdots, t_N^k\}$. Following Zhou et al. (2022b); Li et al. (2024), the $CLASS_n$ context in each $t^k$ is represented by a sequence of class-specific vectors, and the learnable contexts are defined in the word embedding space, then $t^k$ for class $n$ can be formulated as follows:

$$t_n^k = [V]_{1,n}^k ... [V]_{M,n}^k [C_n], \tag{8}$$

where the $M$ denotes the max length of learnable context. The position of $[C_n]$ can also be adjusted. Following Li et al. (2024), we apply the end position as the default position.

For adversarial mixture prompts, We first input different text prompts into the text encoder to obtain individual text features, then we aggregate these text features into a mixture text feature, which can be formulated as follows:

$$z_t^{n,i} = \sum_{k=1}^{K} \tilde{w}_k^i F_{\theta_t}(t_n^k), \tag{9}$$

where $\tilde{w}_k^i$ denotes the weights of adversarial prompt $t^k$ for the adversarial examples $\tilde{x}_i$, and $\tilde{w}_k^i$ is irrelevant to the class (not effected by $n$). $z_t^{n,i}$ denotes the mixture text feature of $n$-th class for the adversarial examples $\tilde{x}_i$. In this way, we can obtain adversarial mixture prompts with pretty diversity through adversarial training to defend against different adversarial examples.

---

**Algorithm 1** Training Process of AMPT

---

**Require:** The train dataset $\mathcal{D}$, clean examples $x$ and adversarial examples $\tilde{x}$, ground truth $y$, text encoder $F_{\theta_t}$ and image encoder $F_{\theta_v}$, adversarial mixture prompts with random initialization $T_m = \{t^1, t^2, \cdots, t^K\}$, total class number $N$, condition prompt weight router $F_{\theta_w}$ with parameter $\theta_w$, the max training epochs $max\text{-}epoch$, the router temperature $\tau_w$

1: **for** 0 to $max\text{-}epoch$ **do**
2:     **for** $Every\ minibatch(x, t, y)\ in\ \mathcal{D}$ **do**
3:         $\tilde{x} = \underset{||\tilde{x}-x|| \le \epsilon}{argmax}\ \mathcal{L}_{AMPT}(\tilde{x}, T_m, y; F_{\theta_v}, F_{\theta_t}, F_{\theta_w})$.
4:         $\{z_{t,1}, \cdots, z_{t,K}\} = \{F_{\theta_t}(t^1), \cdots, F_{\theta_t}(t^K)\}$.
5:         **for** $each\ x_i\ in\ x$ **do**
6:             $\tilde{z}_v^i = F_{\theta_v}(\tilde{x})$.
7:             $\tilde{w}^i = softmax(F_{\theta_w}(\tilde{z}_v^i)/\tau_w)$.
8:             $z_t^{n,i} = \sum_k^K \tilde{w}_k^i z_{t,k}^n$.
9:         **end for**
10:        $\theta_w = \theta_w - \eta \cdot \nabla_{\theta_w} \mathcal{L}_{AMPT}$.
11:        $T_m = T_m - \eta \cdot \nabla_{T_m} \mathcal{L}_{AMPT}$.
12:     **end for**
13: **end for**

---

## 4.3 CONDITIONAL PROMPT WEIGHT ROUTER

Although adversarial mixture prompts can provide diverse adversarial text features, how to select those diverse features still needs to be solved when facing different adversarial examples. Therefore, we focus to adjust the weights: $w_k^i$ in Eq. (9). The simplest approach is to convert $w_k^i$ to $1/K$. To cover diverse image adversarial examples, we propose the conditional prompt weight router, which can generate the image-specific multiple weights for the different adversarial prompts.

Here we design a light-weight network containing two full connection layers as the conditional prompt weight router to predict prompt weights $\tilde{w}^i = \{\tilde{w}_1^i, \cdots, \tilde{w}_K^i\}$ of image $x_i$. Initially, we obtain the adversarial image feature from the image encoder of VLMs $\tilde{z}_v^i$, then we apply the conditional prompt weight router to predict the different weights, which can be formulated as follows:

$$\tilde{w}^i = softmax(F_{\theta_w}(\tilde{z}_v^i)/\tau_w), \tag{10}$$

where $F_{\theta_w}$ denotes the conditional prompt weight router, the $z_v^i$ denotes the feature generated by image encoder of image $x_i$. The softmax operation can keep the sum of the weights as 1. The $\tau_w$ is applied to control the adjustment strength of the generated weight, while a smaller $\tau_w$ denotes stronger adjustment strength, and a larger $\tau_w$ denotes weaker adjustment strength, when $\tau_w$ approaches infinity, it will degenerate into $1/K$. With the assistance of an adaptive weight router, we can finally obtain a more generalizable and representative mixture text feature based on the image features, to further improve the adaptability to defend against different adversarial examples. Meanwhile, we also provide the Theorem 1 about our conditional prompt weight router.

**Theorem 1.** *Assume there are multiple different adversarial text prompts $T_m = \{t^1, t^2, \cdots, t^K\}$, and the corresponding error risk of $k$-th text prompt $t^k$ for adversarial examples $\tilde{x}$ is $\mathcal{R}(\tilde{x}, t^k, y)$, and the normalized prompt weights $\tilde{w} = \{\tilde{w}_1, \tilde{w}_2, \cdots, \tilde{w}_K\}$ are optimized to minimize the error risk expectation of adversarial example $\tilde{x}$, we can obtain:*

$$\mathbb{E}(\sum_k^K \tilde{w}_k \mathcal{R}(\tilde{x}, t^k, y)) \le \mathbb{E}(\frac{1}{K} \sum_k^K \mathcal{R}(\tilde{x}, t^k, y)), \tag{11}$$

*when there exists at least one pair $(i, j)$, $i \ne j$, such that $\mathcal{R}(\tilde{x}, t^i, y)) < \mathcal{R}(\tilde{x}, t^j, y))$, the strict inequality in Eq.(11) holds.*

The proof of Theorem 1 can be viewed in AppendixA.1. Theorem 1 shows that conditional prompt weights can bring the smaller error expectation of the adversarial examples compared with the average error expectation of the adversarial examples, which further demonstrates the necessity and effectiveness of our conditional prompt weight router.

Then the entire process of our AMPT can be viewed in Figure 1, and the optimization loss function $\mathcal{L}_{AMPT}$ can be defined as follows:

$$\mathcal{L}_{AMPT} = -\sum_{n}^{N} y_{in} log \frac{exp(cos(\tilde{z}_v^i, z_t^{n,i}))}{\sum_{n}^{N} exp(cos(\tilde{z}_v^i, z_t^{n,i}))}, \tag{12}$$

and the final training process can be viewed in Algorithm 1. It should be mentioned that to minimize the computational cost, we further decouple Eq. (9) and compute each text feature in advance for a minibatch. For each image, the final mixture text feature is obtained based on pre-computed text features without redundant calculation.

## 5 EXPERIMENTS

### 5.1 EXPERIMENTAL SETTING

***Datasets.*** Following Li et al. (2024), we conduct our experiments on 11 high-resolution vision datasets: ImageNet Deng et al. (2009), Caltech101 Fei-Fei et al. (2004), OxfordPets Parkhi et al. (2012), StanfordCars Krause et al. (2013), Flowers102 Nilsback & Zisserman (2008), Food101 Bossard et al. (2014), FGVCAircraft Maji et al. (2013), SUN397 Xiao et al. (2010), DTD Cimpoi et al. (2014), EuroSAT Helber et al. (2019), and UCF101 Soomro et al. (2012).

***Models.*** Following the setting in Li et al. (2024), we apply ViT-B/32 as our default selected backbone of image encoder, and select the model trained by a strong AT method TeCoA Mao et al. (2022) as our default optimized weight.

***Baselines.*** Because our AMPT is a text prompt tuning method, we mainly compare our method with some similar state-of-the-art methods: Hand Engineered Prompts (HEP, details can be viewed in Appendix A.5), VPT Mao et al. (2022), AdvPT Zhang et al. (2024), APT Li et al. (2024), FAP Zhou et al. (2024), where VPT is a visual prompt tuning method, AdvPT and APT are text prompt tuning method, FAP is the bi-modal tuning method. Here we apply the HEP following the setting in Li et al. (2024). Meanwhile, we change the setting of AdvPT into the setting of APT Li et al. (2024) for the sake of fair comparison. To ensure fairness, we apply the same robust backbone to further enhance the robustness for all the baselines.

***Evaluation Metric.*** Following the setting in Li et al. (2024), we select two adversarial attacks, PGD attack Madry et al. (2017) and AutoAttack Croce & Hein (2020b). If without additional claim, we set the maximum perturbation $\epsilon$ of adversarial attacks to 4/255. For the PGD attack, we apply 100 iterations with a step $\epsilon/4$ following Li et al. (2024). Meanwhile, we employ an ensemble attack, AutoAttack (AA) Croce & Hein (2020b), which consists of four different attack methods: Auto-PGD (APGD), the Difference of Logits Ratio (DLR) attack, FAB-Attack Croce & Hein (2020a), and the black-box Square Attack Andriushchenko et al. (2020). All the methods are evaluated on the entire test test if without additional instruction. For the evaluation of ImageNet against Autoattack, we select the 5000 test set to reduce the calculation overhead following Li et al. (2024), while conducting the AutoAttack on the entire test set is too expensive.

***Training settings.*** For each data set, we perform 16-shot and "all" training, where 16-shot denotes the 16 examples per class randomly sampled from the full training set for model training. As for the training setting of our AMPT, we train all the models with epoch 50 except ImageNet. Due to the high calculation overhead, we train on ImageNet with epoch 20 for "all" shot dataset and apply 100-shot similar to Li et al. (2024). In the maximization of AMPT, we generate the adversarial examples using 3 steps with a step size of $2\epsilon/3$. Meanwhile, we set the prompt length to 16 and the number of prompts of our AMPT to 8 except Sun397, Stanfordcars, and ImageNet. Due to the limitation of computing resources, for Sun397, Stanfordcars, and ImageNet, we set to prompt number as 3. Meanwhile, we set the hyper-parameter $\tau$ as 0.7. The corresponding discussion can be viewed in the Ablation Study. Meanwhile, we conduct the experiments on RTX 4090 except ImageNet, while ImageNet is conducted on A100.

### 5.2 ROBUSTNESS PERFORMANCE

We conduct a benchmark evaluation of our AMPT and baseline approaches. Table 1 and Table 4 present the performance of various prompt methods in 11 datasets in both full-data and 16-shot

Table 1: Robustness performance(%) with all data training setting on 11 different datasets under maximum perturbation 4/255.

| Methods | Metric | ImageNet | Caltech101 | OxfordPets | Flowers102 | Cars | FGVC | DTD | SUN397 | Food101 | EuroSAT | UCF101 | Average |
|---|---|---|---|---|---|---|---|---|---|---|---|---|---|
| HEP | Clean | 39.84 | 77.44 | 61.49 | 30.37 | 10.33 | 7.02 | 27.13 | 31.98 | 21.70 | 20.31 | 36.16 | 33.07 |
| | PGD | 10.27 | 44.02 | 14.28 | 8.73 | 0.92 | 0.48 | 11.17 | 5.86 | 3.19 | 9.25 | 6.24 | 10.40 |
| | AA | 7.24 | 39.92 | 11.01 | 6.41 | 0.62 | 0.06 | 9.52 | 3.94 | 1.76 | 8.21 | 4.84 | 9.50 |
| VPT Mao et al. (2022) | Clean | 48.84 | 84.63 | 62.25 | 67.19 | 1.07 | 0.99 | 22.05 | 48.91 | 39.89 | 78.89 | 12.11 | 42.44 |
| | PGD | 5.78 | 51.36 | 15.51 | 34.51 | 0.91 | 0.99 | 9.99 | 17.48 | 14.15 | 51.70 | 4.57 | 18.81 |
| | AA | 1.44 | 11.52 | 0.05 | 1.79 | 0.65 | 0.00 | 1.77 | 0.51 | 0.49 | 5.26 | 0.32 | 2.16 |
| FAP Zhou et al. (2024) | Clean | 52.17 | 91.03 | 80.07 | 86.43 | 50.21 | 23.88 | 60.81 | 58.35 | 64.38 | 89.71 | 68.25 | 65.94 |
| | PGD | 7.39 | 54.27 | 12.78 | 27.81 | 2.11 | 1.32 | 20.74 | 6.74 | 6.67 | 22.66 | 14.53 | 16.09 |
| | AA | 0.89 | 11.88 | 1.17 | 2.67 | 0.27 | 0.39 | 8.09 | 0.74 | 0.94 | 19.23 | 1.71 | 4.36 |
| AdvPT Zhang et al. (2024) | Clean | 44.60 | 88.88 | 75.58 | 81.49 | 41.47 | 21.66 | 53.78 | 50.34 | 45.42 | 79.58 | 63.44 | 58.75 |
| | PGD | 9.05 | 56.95 | 12.97 | 28.46 | 2.82 | 2.04 | 20.27 | 6.80 | 5.79 | 10.37 | 12.79 | 15.30 |
| | AA | 7.02 | 55.13 | 11.07 | 24.73 | 1.62 | 1.26 | 18.79 | 5.50 | 4.06 | 9.22 | 10.73 | 13.56 |
| APT Li et al. (2024) | Clean | 41.48 | 88.32 | 72.58 | 80.88 | 37.42 | 20.49 | 52.19 | 47.29 | 35.32 | 68.67 | 59.00 | 54.88 |
| | PGD | 12.57 | 63.65 | 24.56 | 44.90 | 8.93 | 7.05 | 26.24 | 13.15 | 13.11 | 24.51 | 21.89 | 23.69 |
| | AA | 8.16 | 61.01 | 16.43 | 38.61 | 3.92 | **3.33** | 22.40 | 8.06 | 7.32 | 29.79 | 16.39 | 19.58 |
| **AMPT (ours)** | Clean | 42.30 | 87.38 | 72.72 | 82.34 | 45.17 | 20.58 | 53.43 | 51.48 | 38.98 | 68.19 | 60.27 | 56.62 |
| | PGD | **12.62** | **65.03** | **24.78** | **46.81** | **12.16** | **7.56** | **28.49** | 13.92 | 13.34 | 37.70 | **21.99** | **25.85** |
| | AA | **8.18** | **62.39** | **16.57** | **41.21** | **6.01** | 3.21 | **25.41** | **9.91** | **8.03** | **34.56** | **17.29** | **21.16** |

training settings. Based on the results, AMPT improves robustness by an average of 8.99% and 11.33% (PGD/AA) in the all-shot setting, and by 8.60% and 8.56% (PGD/AA) in the 16-shot setting. Furthermore, AMPT achieves an average accuracy improvement of 5.60% and 8.55% (under all /16 shots training settings). It demonstrates strong adversarial robustness across various attacks while maintaining competitive accuracy.

Specifically, AMPT consistently outperforms AdvPT and the best baseline APT in robustness and data efficiency. Under full-data training, AMPT improves robustness over the best baseline by 2.16% and 1.58% (PGD/AA). In the 16-shot setting, AMPT surpasses the baseline by 0.68% and 0.87% (PGD/AA), demonstrating its enhancement of APT's performance under various attacks. In contrast to AMPT, VPT and FAP perform poorly in evaluating AA attacks, likely due to their lack of the generalization ability to unseen attacks, as seen in AMPT.

## 5.3 GENERALIZATION ACROSS DIFFERENT DATASETS

Meanwhile, we test the generalization across different datasets. We apply the APT and AMPT trained on Caltech101 with all-data training setting as source models and evaluate the adversarial robustness in different datasets, including Oxford-Pets, OxfordFlowers, StanfordCars, FGVCAir-

Table 2: Comparison of robust generalization (%) between APT and AMPT across 9 datasets.

| Metric | Clean | PGD | AA |
|---|---|---|---|
| APT | 14.75 | 3.39 | 2.66 |
| **AMPT (ours)** | **18.39** | **4.89** | **4.14** |

craft, Sun397, DTD, Food101, EuroSAT, and UCF101, and the average results are reported in Table 2. From the result, we find that AMPT has better robust generalization compared with APT. Specifically, AMPT has a 1.5% and 1.48% robustness improvement against PGD and AA attacks compared with APT, showing that AMPT has better performance in dealing with diverse adversarial examples even in unseen classes. The detailed results can be viewed in Appendix A.2.

## 5.4 ABLATION STUDY

To verify the effectiveness of AMPT, we conduct a set of ablation studies. We conduct the experiment in the Caltech101 dataset with the 16-shot training setting.

Table 3: Ablation Study towards different components.

| Component | Clean | PGD | AA |
|---|---|---|---|
| Baseline | 86.29 | 56.75 | 53.43 |
| Baseline+Mixture | 87.06 | 57.36 | 54.60 |
| Baseline+Mixture+Router | **87.14** | **57.69** | **55.05** |

***Effects of Different Components.*** We conduct ablation studies on different components. Starting from the single adversarial cue fine-tuning baseline, we first add adversarial mixture prompts, and then

Table 4: Robustness performance(%) with 16-shot training setting on 11 different datasets under maximum perturbation 4/255.

| Methods | Metric | ImageNet | Caltech101 | OxfordPets | Flowers102 | Cars | FGVC | DTD | SUN397 | Food101 | EuroSAT | UCF101 | Average |
|---|---|---|---|---|---|---|---|---|---|---|---|---|---|
| HEP | Clean | 39.84 | 77.44 | 61.49 | 30.37 | 10.33 | 7.02 | 27.13 | 31.98 | 21.70 | 20.31 | 36.16 | 33.07 |
| | PGD | 10.27 | 44.02 | 14.28 | 8.73 | 0.92 | 0.48 | 11.17 | 5.86 | 3.19 | 9.25 | 6.24 | 10.40 |
| | AA | 7.24 | 39.92 | 11.01 | 6.41 | 0.62 | 0.06 | 9.52 | 3.94 | 1.76 | 8.21 | 4.84 | 9.50 |
| VPT Mao et al. (2022) | Clean | 34.84 | 76.92 | 3.38 | 41.49 | 3.38 | 1.05 | 11.64 | 44.02 | 1.16 | 6.70 | 2.11 | 20.61 |
| | PGD | 3.13 | 28.28 | 0.25 | 13.85 | 0.25 | 0.93 | 0.71 | 13.39 | 0.08 | 0.00 | 0.13 | 5.55 |
| | AA | 0.71 | 0.30 | 0.14 | 0.20 | 0.14 | 0.00 | 0.24 | 0.30 | 0.09 | 0.10 | 0.19 | 0.22 |
| FAP Zhou et al. (2024) | Clean | 50.34 | 89.85 | 76.09 | 76.24 | 43.68 | 19.44 | 50.29 | 56.24 | 55.39 | 64.67 | 63.75 | 58.73 |
| | PGD | 6.92 | 49.77 | 11.28 | 17.45 | 1.67 | 1.32 | 16.72 | 2.44 | 4.21 | 13.48 | 10.46 | 12.34 |
| | AA | 0.51 | 8.92 | 1.74 | 1.46 | 0.18 | 0.21 | 1.40 | 0.66 | 11.17 | 0.97 | 3.10 |
| AdvPT Zhang et al. (2024) | Clean | 43.09 | 87.58 | 73.29 | 74.46 | 37.07 | 19.92 | 46.45 | 47.28 | 36.05 | 61.40 | 56.01 | 52.96 |
| | PGD | 8.72 | 50.67 | 12.84 | 20.99 | 2.69 | 2.07 | 16.13 | 6.45 | 4.31 | 9.07 | 10.52 | 13.13 |
| | AA | 6.72 | 49.53 | 10.47 | 16.89 | 1.69 | 0.96 | 14.54 | 5.17 | 2.99 | 7.36 | 9.09 | 11.40 |
| APT Li et al. (2024) | Clean | 41.12 | 86.29 | 67.29 | 76.41 | 31.6 | 20.31 | 45.86 | 44.92 | 30.39 | 64.33 | 53.16 | 51.06 |
| | PGD | 12.27 | 56.75 | 19.98 | 37.52 | 7.70 | 6.15 | 21.51 | 10.94 | 7.90 | **25.54** | 16.55 | 20.26 |
| | AA | **7.88** | 53.43 | **13.46** | 32.20 | 3.37 | **2.64** | 18.91 | 6.88 | 4.17 | 16.68 | 12.74 | 15.67 |
| **AMPT (ours)** | Clean | 41.20 | 87.14 | 70.62 | 76.53 | 38.54 | 19.29 | 47.75 | 47.32 | 30.73 | 57.06 | 54.09 | 51.84 |
| | PGD | **12.38** | **57.69** | **20.69** | **37.79** | **9.29** | **6.63** | **22.99** | 11.08 | **8.46** | 24.72 | **18.61** | **20.94** |
| | AA | 7.84 | **55.05** | 13.27 | **32.28** | **5.07** | 2.52 | **20.04** | **7.75** | **4.60** | **19.35** | 14.14 | **16.54** |

further incorporate the conditional prompt weight router. Table 3 reports the results on Caltech101, with others given in Appendix A.3.

The results verify the effectiveness of each component in AMPT: adding mixture prompts without the weight router yields a 2.44% improvement over the baseline against PGD attack, while incorporating the conditional prompt weight router provides a 0.93% gain. This demonstrates that the feature diversity brought by adversarial mixture prompts and the adaptability brought by the conditional prompt weight router work together to improve the generalization and robustness of VLMs.

***Selection of Prompt number.*** We explore the selection of prompt numbers. We select the following text prompt number of our AMPT as 1, 2, 4, 6, 8, 10, 12, and the result can be viewed in Table 5. From the results, when the number of prompts increases at the beginning (from number 1 to 8), the adversarial robustness of AMPT will obviously increase. However, when it further increases (from number 8 to 12), the robustness remains basically unchanged. It can be explained that as the number of prompts increases, the difficulty of prompt optimization also increases. Thus, we select the prompt number 8 as the default setting.

Table 5: Ablation on Prompt Number

| Prompt number | Clean | PGD | AA |
|---|---|---|---|
| 1 | 86.29 | 56.75 | 53.43 |
| 2 | 86.13 | 56.98 | 53.66 |
| 4 | 86.69 | 57.45 | 54.52 |
| 6 | 87.22 | 57.04 | 54.32 |
| **8** | 87.14 | **57.69** | **55.05** |
| 10 | 87.34 | 56.80 | 54.40 |
| 12 | 87.55 | 57.20 | 54.44 |

***Selection of Hyper-parameter*** $\tau$**.** The temperatures $\tau$ can control the adjustment strength of the conditional prompt weight router. While smaller $\tau$ means larger adjustment strength, larger $\tau$ means smaller adjustment strength. We select the following $\tau$ of AMPT as 0.3, 0.5, 0.7, 0.9, and 1.1, and the results can be found in Table 6. Based on the experimental results, we select the Hyper-parameter $\tau$ to 0.7.

Table 6: Ablation on Hyper-parameter $\tau$.

| $\tau$ | Clean | PGD | AA |
|---|---|---|---|
| 0.3 | 85.72 | 55.98 | 53.18 |
| 0.5 | 86.86 | 57.93 | 54.32 |
| **0.7** | 87.14 | 57.69 | **55.05** |
| 0.9 | 87.14 | 57.32 | 54.69 |
| 1.1 | 87.05 | **57.77** | 54.56 |

## 6 CONCLUSION

In this work, we focused on the overfitting problem of adversarial prompt tuning, and found that simply increasing the length of the text prompt led to the learning difficulty while increasing the number of prompts was more likely to improve the adversarial robustness of the VLMs. Based on the observation, we propose Adversarial Mixture Prompt Tuning (AMPT), which introduces adversarial mixture prompts to obtain more general text features, and proposes a conditional prompt weight router to further improve the adaptability of adversarial mixture prompts. Our theoretical analysis validates the effectiveness of the router. Extensive experiments demonstrate that AMPT consistently improves in-distribution robustness and exhibits strong transfer robustness across diverse datasets.

ETHICS STATEMENT

In this study, we fully adhere to the ethical standards of academic research, particularly in the use of datasets, algorithms, and models. All experiments are conducted using publicly available datasets, and we strictly comply with the usage permissions and related regulations provided by the data providers, ensuring that no human or sensitive data is involved. Additionally, all experiments follow the principles of fairness, transparency, and verifiability, with the aim of advancing technology without causing negative societal impacts.

REPRODUCIBILITY STATEMENT

To ensure the reproducibility of our findings, we have provided the core code for AMPT in the supplementary material, allowing other researchers to replicate our work under the same conditions. Our experimental environment uses model configurations commonly used in academia and clearly documents the hardware configuration and hyper-parameters (in Section 5.1). The pseudocode for the proposed AMPT is given in Algorithm 1.

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

## A  APPENDIX

### A.1  THE PROOF OF THEOREM 1

Here we assume each text prompt keeps unchanged and only optimize the prompt weights $\tilde{w} = \{\tilde{w}_1, \tilde{w}_2, \cdots, \tilde{w}_K\}$ from initial weights $\tilde{w}^0 = \{1/K, 1/K, \cdots, 1/K\}$. Then as for any adversarial example $\tilde{x}'$, without loss generalization, we can assmue a corresponding text prompt $t^k$ exist error risk as follows:

$$0 \le \mathcal{R}(\tilde{x}', t^k, y) \le \mathcal{R}(\tilde{x}', t^1, y) \le \cdots \le \mathcal{R}(\tilde{x}', t^{k-1}, y)$$
$$\le \mathcal{R}(\tilde{x}', t^{k+1}, y) \le \cdots \le \mathcal{R}(\tilde{x}', t^K, y). \tag{13}$$

Thus based on the Gradient Descent operation, the expectation of corresponding weights can be formulated as follows:

$$\mathbb{E}(\tilde{w}'_k) = \frac{1}{K} - \eta \cdot \frac{\partial \frac{1}{K} \sum_k^K \mathcal{R}(\tilde{x}', t^k, y)}{\partial \tilde{w}_k}, \tag{14}$$

and here we assume that different weights are independent of each other, so we can obtain the $\mathbb{E}(\tilde{w}'_K)$ as follows:

$$\mathbb{E}(\tilde{w}'_k) = \frac{1}{K} - \eta \cdot \mathcal{R}(\tilde{x}', t^k, y). \tag{15}$$

Then, we apply softmax to each $\mathbb{E}(\tilde{w}'_k)$, and denote the result of applying softmax to $\mathbb{E}(\tilde{w}'_k)$ is $\mathbb{E}(\tilde{w}_k)$, then we get:

$$\mathbb{E}(\tilde{w}'_k) = \frac{e^{\frac{1}{K} - \eta \cdot \mathcal{R}(\tilde{x}', t^k, y)}}{\sum_j^K e^{\frac{1}{K} - \eta \cdot \mathcal{R}(\tilde{x}', t^j, y)}}. \tag{16}$$

For simplicity, we let $\mathcal{R}_k = \mathcal{R}(\tilde{x}', t^k, y)$, so Eq. (16)It can be further simplified to:

$$\mathbb{E}(\tilde{w}_k) = \frac{e^{-\eta \cdot \mathcal{R}_k}}{\sum_j^K e^{-\eta \cdot \mathcal{R}_j}}. \tag{17}$$

Then the entire error expectation of adversarial examples can be formulated as follows:

$$\mathbb{E}(\sum_k^K \tilde{w}_k \mathcal{R}_k) = \mathbb{E}(\frac{1}{K} \sum_k^K \mathcal{R}_k) + \sum_k^K (\mathbb{E}(\tilde{w}_k) - \frac{1}{K}) * \mathcal{R}_k. \tag{18}$$

After substituting $\mathbb{E}(\tilde{w}_k)$ into $\sum_k^K (\mathbb{E}(\tilde{w}_k) - \frac{1}{K}) * \mathcal{R}_k$, we have:

$$\sum_k^K (\mathbb{E}(\tilde{w}_k) - \frac{1}{K}) * \mathcal{R}_k = \sum_k^K (\frac{e^{-\eta \cdot \mathcal{R}_k}}{\sum_j^K e^{-\eta \cdot \mathcal{R}_j}} - \frac{1}{K}) \cdot \mathcal{R}_k. \tag{19}$$

We assume the following relation holds:

$$\sum_k^K (\frac{e^{-\eta \cdot \mathcal{R}_k}}{\sum_j^K e^{-\eta \cdot \mathcal{R}_j}} - \frac{1}{K}) \cdot \mathcal{R}_k \le 0, \tag{20}$$

the above inequality is equivalent to the following inequality:

$$\sum_k^K (e^{-\eta \cdot \mathcal{R}_k} - \frac{\sum_j^K e^{-\eta \cdot \mathcal{R}_j}}{K}) \cdot \mathcal{R}_k \le 0. \tag{21}$$

When $K = 2$, without loss of generality, we assume that $\mathcal{R}_1 \le \mathcal{R}_2$, then we have:

$$\sum_k^K (e^{-\eta \cdot \mathcal{R}_k} - \frac{\sum_j^K e^{-\eta \cdot \mathcal{R}_j}}{K}) \cdot \mathcal{R}_k$$
$$= \frac{e^{-\eta \cdot \mathcal{R}_1} - e^{-\eta \cdot \mathcal{R}_2}}{2} \cdot (\mathcal{R}_1 - \mathcal{R}_2). \tag{22}$$

Since $\mathcal{R}_1 \leq \mathcal{R}_2$, So we have $\mathcal{R}_1 - \mathcal{R}_2 \leq 0$ and $\frac{e^{-\eta \cdot \mathcal{R}_1} - e^{-\eta \cdot \mathcal{R}_2}}{2} \geq 0$, which means when $K = 2$, the assumption holds. Next, we assume that the assumption holds when $K = n$, and when $K = n + 1$, without loss of generality, suppose $\mathcal{R}_{n+1}$ satisfies:

$$\mathcal{R}_{n+1} \leq \mathcal{R}_1 \leq \cdots \leq \mathcal{R}_n, \tag{23}$$

then we have:

$$\sum_{k}^{n+1} (e^{-\eta \cdot \mathcal{R}_k} - \frac{\sum_{j}^{n+1} e^{-\eta \cdot \mathcal{R}_j}}{K}) \cdot \mathcal{R}_k$$

$$= \sum_{k}^{n+1} e^{-\eta \cdot \mathcal{R}_k} \cdot \mathcal{R}_k - \frac{\sum_{j}^{n+1} e^{-\eta \cdot \mathcal{R}_j}}{n+1} \cdot \sum_{k}^{n+1} \cdot \mathcal{R}_k. \tag{24}$$

By the induction hypothesis, we have:

$$\sum_{k}^{n+1} e^{-\eta \cdot \mathcal{R}_k} \cdot \mathcal{R}_k = \sum_{k}^{n} e^{-\eta \cdot \mathcal{R}_k} \cdot \mathcal{R}_k + e^{-\eta \cdot \mathcal{R}_{n+1}} \cdot \mathcal{R}_{n+1}$$

$$\leq \frac{\sum_{j}^{n} e^{-\eta \cdot \mathcal{R}_j}}{n} \cdot \sum_{k}^{n} \mathcal{R}_k + e^{-\eta \cdot \mathcal{R}_{n+1}} \cdot \mathcal{R}_{n+1}, \tag{25}$$

meanwhile, we have:

$$\frac{\sum_{j}^{n+1} e^{-\eta \cdot \mathcal{R}_j}}{n+1} \cdot \sum_{k}^{n+1} \cdot \mathcal{R}_k$$

$$= \frac{\sum_{j}^{n} e^{-\eta \cdot \mathcal{R}_j}}{n+1} \cdot \sum_{k}^{n} \mathcal{R}_k + \frac{\sum_{j}^{n} e^{-\eta \cdot \mathcal{R}_j}}{n+1} \cdot \mathcal{R}_{n+1}$$

$$+ \frac{e^{-\eta \cdot \mathcal{R}_{n+1}}}{n+1} \cdot \sum_{k}^{n} \mathcal{R}_k + \frac{e^{-\eta \cdot \mathcal{R}_{n+1}}}{n+1} \cdot \mathcal{R}_{n+1}. \tag{26}$$

Substituting Eq. (26) and Eq. (25) into Eq. (24), we can obtain:

$$\sum_{k}^{n+1} (e^{-\eta \cdot \mathcal{R}_k} - \frac{\sum_{j}^{n+1} e^{-\eta \cdot \mathcal{R}_j}}{n+1}) \cdot \mathcal{R}_k$$

$$\leq [(\frac{\sum_{j}^{n} e^{-\eta \cdot \mathcal{R}_j}}{n(n+1)} \cdot \sum_{k}^{n} \mathcal{R}_k - \frac{e^{-\eta \cdot \mathcal{R}_{n+1}}}{n+1} \cdot \sum_{k}^{n} \mathcal{R}_k)$$

$$+ (\frac{n \cdot e^{-\eta \cdot \mathcal{R}_{n+1}}}{n+1} \cdot \mathcal{R}_{n+1} - \frac{\sum_{j}^{n} e^{-\eta \cdot \mathcal{R}_j}}{n+1} \cdot \mathcal{R}_{n+1})]$$

$$= \frac{1}{n+1} \cdot (\frac{\sum_{j}^{n} e^{-\eta \cdot \mathcal{R}_j}}{n} - e^{-\eta \cdot \mathcal{R}_{n+1}}) \cdot (\sum_{k}^{n} \mathcal{R}_k - n \cdot \mathcal{R}_{n+1}).$$

Since we have previously assumed that $\mathcal{R}_{n+1} \leq \mathcal{R}_1 \leq \cdots \leq \mathcal{R}_n$, so we can deduce that:

$$\sum_{k}^{n} \mathcal{R}_k - n \cdot \mathcal{R}_{n+1} \geq 0, \tag{27}$$

and:

$$\frac{\sum_{j}^{n} e^{-\eta \cdot \mathcal{R}_j}}{n} - e^{-\eta \cdot \mathcal{R}_{n+1}} \leq 0. \tag{28}$$

Therefore, we can conclude that:

$$\sum_{k}^{n+1} (e^{-\eta \cdot \mathcal{R}_k} - \frac{\sum_{j}^{n+1} e^{-\eta \cdot \mathcal{R}_j}}{K}) \cdot \mathcal{R}_k \leq 0. \tag{29}$$

Table 7: Detailed results comparing the Robust Generalization (%) between APT and AMPT cross 9 different datasets based on Caltech101 adversarial prompts.

| Method | Metric | OxfordPets | Flowers102 | Cars | FGVC | DTD | SUN397 | Food101 | EuroSAT | UCF101 | Average |
|--------|--------|-----------|-----------|------|------|-----|--------|---------|---------|--------|---------|
| APT | Clean | 29.95 | 14.01 | 8.10 | 1.83 | **16.84** | 14.19 | **15.52** | **12.81** | 19.51 | 14.75 |
| | PGD | **10.06** | 3.17 | 0.87 | 0.36 | **7.74** | 2.25 | 1.43 | 1.14 | 3.49 | 3.39 |
| | AA | **8.67** | 2.15 | 0.39 | 0.33 | 6.74 | 1.63 | 0.77 | 0.58 | 2.72 | 2.66 |
| **AMPT(ours)** | Clean | **45.08** | **16.00** | **13.01** | **2.34** | 16.02 | **20.17** | 15.3 | 11.43 | **26.20** | **18.39** |
| | PGD | 9.10 | **4.02** | **1.02** | **0.81** | 7.68 | **4.10** | **2.24** | **10.67** | **4.34** | **4.89** |
| | AA | 7.14 | **2.84** | **0.53** | **0.75** | **6.97** | **4.02** | **1.31** | **10.57** | **3.09** | **4.14** |

Table 8: Detailed ablation results cross 10 different datasets.

| Method | Metric | Caltech101 | OxfordPets | Flowers102 | Cars | FGVC | DTD | SUN397 | Food101 | EuroSAT | UCF101 | Average |
|--------|--------|-----------|-----------|-----------|------|------|-----|--------|---------|---------|--------|---------|
| Baseline | Clean | 86.29 | 67.29 | 76.41 | 31.6 | **20.31** | 45.86 | 44.92 | 30.39 | **64.33** | 53.16 | 52.06 |
| | PGD | 56.75 | 19.98 | 37.52 | 7.7 | 6.15 | 21.51 | 10.94 | 7.9 | **25.54** | 16.55 | 21.05 |
| | AA | 53.43 | 13.46 | 32.20 | 3.37 | **2.64** | 18.91 | 6.88 | 4.17 | 16.68 | 12.74 | 16.45 |
| Baseline+Mixture | Clean | 86.45 | 69.61 | 76.45 | 34.56 | 17.88 | 46.39 | 45.51 | 30.2 | 56.89 | **55.22** | 51.92 |
| | PGD | 57.24 | 20.66 | 37.61 | 7.8 | 6.21 | 21.93 | 11.01 | 8.14 | 24.3 | 18.55 | 21.35 |
| | AA | 53.75 | 14.01 | 32.23 | 3.55 | 2.49 | 18.79 | 6.89 | 4.38 | 15.53 | 13.35 | 16.5 |
| Baseline+Mixture+Router | Clean | **87.14** | **70.62** | **76.53** | **38.54** | 19.29 | **47.75** | **47.32** | **30.73** | 57.06 | 54.09 | **52.91** |
| | PGD | **57.69** | **20.69** | **37.79** | **9.29** | **6.63** | **22.99** | **11.08** | **8.46** | 24.72 | **18.61** | **21.8** |
| | AA | **55.05** | **13.27** | **32.28** | **5.07** | 2.52 | **20.04** | **7.75** | **4.6** | 19.35 | **14.14** | **17.41** |

Therefore, by mathematical induction, we can conclude that the following holds for any K:

$$\sum_{k}^{K} (e^{-\eta \cdot \mathcal{R}_k} - \frac{\sum_{j}^{K} e^{-\eta \cdot \mathcal{R}_j}}{K}) \cdot \mathcal{R}_k \leq 0. \tag{30}$$

Substituting Eq. (30) into Eq. (18), we obtain:

$$\mathbb{E}(\sum_{k}^{K} \tilde{w}_k \mathcal{R}_k) \leq \mathbb{E}(\frac{1}{K} \sum_{k}^{K} \mathcal{R}_k). \tag{31}$$

Furthermore, form the above proof, we can deduce that if there exists at least one pair $(i, j)$, $i \neq j$, such that $\mathcal{R}_i < \mathcal{R}_j$, we have:

$$\mathbb{E}(\sum_{k}^{K} \tilde{w}_k \mathcal{R}_k) < \mathbb{E}(\frac{1}{K} \sum_{k}^{K} \mathcal{R}_k). \tag{32}$$

Then the Theorem 1 is proved.

## A.2 DETAILS ON GENERALIZATION CROSS DIFFERENT DATASETS

As shown in Table 7, we present the robust generalization performance across different target datasets when trained on Caltech101 as the source domain. The results show that our AMPT consistently outperforms APT in adversarial robustness across 8 datasets, and achieves higher accuracy on 6 datasets, indicating superior generalization ability and enhanced resilience against diverse attacks.

## A.3 DETAILS ON ABLATION STUDY

As shown in Table 8, we report ablation results on 10 target datasets. The results indicate that incorporating both mixture prompts and the router consistently improves performance across almost all datasets, with average gains of 0.85/0.75/0.96 under clean accuracy, PGD, and AA attacks, respectively. In contrast, removing the router leads to lower performance than the full combination, suggesting that the two components are complementary: together they maintain high accuracy while enhancing adversarial robustness.

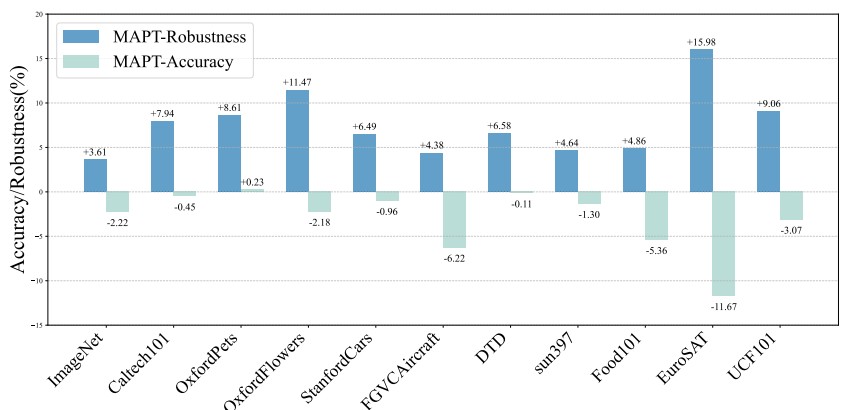

Figure 3: Trade-off between Accuracy and Robustness ($M = 16$).

Table 9: Datasets statistics and their prompts

| Dataset | Classes | Hand-Engineered Prompt |
|---|---|---|
| ImageNet | 1000 | a photo of a [CLASS] |
| Caltech101 | 100 | a photo of a [CLASS] |
| OxfordPets | 37 | a photo of a [CLASS], a type of pet |
| Flowers102 | 102 | a photo of a [CLASS], a type of flower |
| StanfordCars | 196 | a photo of a [CLASS] |
| FGVC | 100 | a photo of a [CLASS], a type of aircraft |
| DTD | 47 | [CLASS] texture |
| SUN397 | 397 | a photo of a [CLASS] |
| Food101 | 101 | a photo of a [CLASS], a type of food |
| EuroSAT | 10 | a centered satellite photo of a [CLASS] |
| UCF101 | 101 | a photo of a person doing [CLASS] |

## A.4 TRADE-OFF BETWEEN ACCURACY AND ROBUSTNESS

As shown in Fig. 3, we compare the performance improvement per dataset of our adversarially-trained prompt over the standard-trained prompt for unified context. Most adversarially trained vision models tend to improve robustness at the expense of accuracy, and adversarially trained prompts also exhibit this trade-off, which is expected. More importantly, we observe that for most datasets, the gain in robustness outweighs the drop in accuracy. Specifically, AMPT improves adversarial robustness by an average of +7.60%, while incurring only a modest drop of -3.03% in accuracy. For instance, on OxfordPets, robustness increases significantly by +8.61%, with a slight gain of +0.23% in accuracy. These results suggest that our method achieves a relatively favorable trade-off between accuracy and robustness.

## A.5 DETAILS OF THE DATASET AND HAND-ENGINEERED PROMPT

The 11 datasets were selected to establish a comprehensive benchmark, covering a wide range of vision tasks including generic object classification, scene recognition, action classification, fine-grained recognition, texture recognition, and satellite imagery analysis. They were split into training and test sets following the protocol of Zhou et al. (2022b). Table 9 reports the number of categories in each dataset along with the corresponding inputs for the hand-engineered prompts.

## A.6 LIMITATION

From the perspective of computational cost, AMPT remains a parameter-efficient and highly competitive method, capable of rapid optimization, as shown in Table 10. The inference memory and time costs of AMPT are slightly higher than those of APT but are still significantly lower than those of FAP, indicating that it maintains high inference efficiency while ensuring robustness. However, due

to the presence of mixture prompts, the training cost of AMPT remains higher than that of APT. In addition, similar to the other parameter-efficient method, AMPT relies on pre-trained VLMs with initial adversarial robustness as claimed in Li et al. (2024), which needs to be further explored in the future.

Table 10: Calculation Overhead between Different Methods. The results are conducted based on RTX 4090 in 16-shot setting of each epoch with Caltech101.

| Method | VPT | APT | FAP | AMPT |
|---|---|---|---|---|
| Training Memory Cost | 6730M | 2798M | 4204M | 14384M |
| Training Time Cost | 30s | 14s | 165s | 60s |
| Inference Memory Cost | 2246M | 5626M | 7478M | 5838M |
| Inference Time Cost | 5.00s | 8.42s | 14.01s | 10.66s |

## A.7 THE USE OF LARGE LANGUAGE MODELS (LLMS)

This study improves upon existing methods without using large language models. Additionally, we relied on traditional linguistic tools for paper refinement.

