# OpenReview forum: "Enhancing Adversarial Robustness of Vision Language Models via Adversarial Mixture Prompt Tuning"
_ICLR.cc/2026/Conference — ICLR 2026 Conference Withdrawn Submission_

### Official Review · Reviewer_AQpm · 2025-10-26

**Soundness:** 3
**Presentation:** 2
**Contribution:** 2
**Rating:** 4
**Confidence:** 2

**Summary:**

This paper proposes Adversarial Mixture Prompt Tuning (AMPT), a parameter-efficient method to enhance the adversarial robustness of Vision-Language Models (VLMs), such as CLIP. Unlike prior adversarial prompt tuning (APT) approaches that optimize a single learnable text prompt, AMPT introduces multiple learnable prompts and a conditional weight router that dynamically combines them based on the adversarial image features. Experiments on 11 datasets under PGD and AutoAttack show improved robustness and generalization over baselines including APT, AdvPT, and FAP.

**Strengths:**

1. The idea of using multiple learnable prompts and dynamically weighting them via an adaptive router is both intuitive and technically sound.
2. The effectiveness of the conditional weight router is theoretically verified, providing a solid justification that adaptive weighting can reduce the expected adversarial error.
3. Experimental results demonstrate consistent and substantial improvements of AMPT over state-of-the-art baselines such as APT and AdvPT in both clean and robust accuracy across almost all datasets, demonstrating the method's effectiveness.
4. The paper is well-organized and easy to follow.

**Weaknesses:**

1. The experiments are not sufficient to fully demonstrate the effectiveness of AMPT:
-  In Section 3.2, the authors show that increasing the number of prompts can further improve robustness compared with increasing the prompt length. However, to more convincingly support this claim, Table 1 should include comparison results with APT under the same total number of prompt parameters, ensuring a fair comparison between the two settings.
- While APT, AdvPT, and FAP are included as baselines, the paper lacks comparisons with Test-Time Adversarial Prompt Tuning (TAPT) (Wang et al., 2025) under the same experimental setup. Including TAPT would provide a more comprehensive evaluation against recent state-of-the-art methods.
- The proposed method relies on a pre-trained VLM backbone. However, the paper does not explore how AMPT’s performance varies or degrades when different backbone models are used, which limits understanding of its robustness transferability and general applicability across architectures.
-  Although the generalization across different datasets are evaluated, the paper does not include a distribution shift test (as conducted in Section 5.2 of APT), which is important for assessing the model’s robustness to domain or input distribution variations.
- Finally, the paper lacks quantitative illustrations or concrete examples of the learned prompts. It remains unclear what specific prompts are obtained through AMPT and how they differ from those generated by baseline methods such as APT. Including representative prompt examples or visualizations would enhance interpretability and transparency.
2. Significant computational overhead: As shown in Table 10, the training cost of AMPT remains considerably higher than that of APT, raising concerns about its practicality for large-scale or real-time applications.

**Questions:**

1. Theorem 1 (Sec. 4.3) claims that the conditional prompt weight router reduces the expected error risk compared to uniform weighting. However, the proof in Appendix A.1 assumes that the risk of each prompt is static and independent of the weight optimization process. In AMPT’s joint training setting, the prompt embeddings and the conditional weights are optimized simultaneously, meaning that each prompt’s risk may depend on the weight updates. Could the authors clarify whether the theoretical guarantee of Theorem 1 still applies under joint training, or is it only valid in the fixed-prompt setting?
2. How sensitive is AMPT to the number of mixture prompts (K) across different datasets? Would adaptive K selection improve further?

---

### Official Review · Reviewer_Bv1L · 2025-10-28

**Soundness:** 3
**Presentation:** 3
**Contribution:** 2
**Rating:** 4
**Confidence:** 4

**Summary:**

This paper addresses the adversarial robustness problem of Vision-Language Models via adversarial prompt tuning. The authors empirically find that, rather than using a single learnable text prompt, training multiple short prompts and mixing them achieves stronger generalization. Based on this observation, the paper proposes Adversarial Mixture Prompt Tuning (AMPT), which learns mixture text prompts with a routing module that adaptively weights the text prompts. Experiments using CLIP ViT-B/32 on 11 image classification datasets show that the proposed method outperforms existing baselines.

**Strengths:**

- Combining short adversarial prompts with a routing module is novel yet simple idea.
- Empirical validation across multiple datasets confirms consistent improvement compared to baselines.

**Weaknesses:**

- The experiments are limited to the CLIP ViT-B/32 model. This is a considerable weakness that limits the generalizability of the claim.
- The routing module introduces additional parameters, so part of the performance improvement may simply stem from the increased model capacity.
- It appears that a long single text prompt is not suitable for a relatively weak model like CLIP ViT-B/32, which is known to struggle with long text sequences. [Zhang+2024] have shown that CLIP’s performance saturates when the input length exceeds around 20 tokens, indicating that using a 64-token prompt is simply beyond the effective capacity of such a small model. The authors should conduct additional experiments with CLIP ViT-L/14, Long-CLIP, or other recent VLMs that better handle extended text inputs.
- Considering that the number of parameters and training cost are larger than in APT, the observed improvement of only 0–1% appears relatively minor.


[Zhang+2024] Zhang, Beichen, et al. "Long-clip: Unlocking the long-text capability of clip." European conference on computer vision. Cham: Springer Nature Switzerland, 2024.

**Questions:**

see weaknesses

---

### Official Review · Reviewer_HDVm · 2025-10-30

**Soundness:** 2
**Presentation:** 2
**Contribution:** 2
**Rating:** 4
**Confidence:** 3

**Summary:**

This paper proposes Adversarial Mixture Prompt Tuning (AMPT) to enhance adversarial robustness of Vision-Language Models (VLMs). The authors first observe that using multiple short prompts outperforms a single long prompt (with equal total parameters) for adversarial robustness. Building on this, AMPT learns K adversarial text prompts and uses a conditional router network to predict sample-specific mixture weights based on adversarial image features. The method is evaluated on 11 datasets against PGD and AutoAttack, showing improvements over baselines.

**Strengths:**

. Finding that multiple short prompts outperform a single long prompt with the same parameter count is an interesting core observation.

. The comprehensive experimental scope that covers 11 datasets demonstrates that the method works across different scenarios.

. While some gains are modest, they are consistently positive, suggesting the approach has merit.

. The method is practical for deployment scenarios where computational resources are limited.

. Unlike many adversarial training methods, the method doesn't sacrifice clean accuracy.

**Weaknesses:**

. You compare K=4 short prompts vs K=1 long prompt and say “more prompts > longer prompt.” But this is really an ensemble (K=4) vs a single model (K=1). Ensembles usually win because of diversity, even with the same total tokens.
Please add a plain ensemble baseline, I mean train K independent APT models (same total tokens), and average the predictions. Then we can see if your mixture is better than a standard ensemble, or not.

. Your ablations show the router adds only ~0.33% over a uniform mixture (and sometimes 0%). With no error bars or tests, this can be noise. Most gain seems to come from more prompts, not from the router. I suggest you add mean and std over multiple seeds and a significance test. My question is, why is the router needed if the gain is this small?

. I think Theorem 1 is too trivial for your method since the theorem says that optimized weights are better than uniform. Also, the proof assumes independent weights, while your weights are softmax‑coupled. It does not show that your two‑layer router can learn good weights and generalize. I suggest removing the theorem and presenting only the intuition for it.

. Inconsistent experimental choices, such as different K across datasets. τ = 0.7 is employed everywhere, but the choice is not discussed. A different protocol is used for ImageNet but not with a full justification.

. Recent works have shown that the test-time defences mostly fail under defence-aware attacks. Your router is a key step and should be targeted. For example, router‑aware adaptive attacks will try to push the router to bad mixtures.

. You claim that prompts play "unique roles," but you don't prove this. You should measure the similarities among learned prompts, demonstrate leave‑one‑prompt‑out impact, and provide a visualization of router weights for both clean and adversarial images.

. Some terms are used without clear definitions, such as "unique roles" and "on‑the‑fly".

. Only CLIP ViT‑B/32 (with TeCoA backbone) is tested; generalization to other VLMs/backbones should be tested.

. You train on Caltech101 but test on other datasets; prompts are per‑class in (Eq. 8). How do you apply them when classes differ? please explain the cross‑dataset procedure.

**Questions:**

Please check weaknesses.

---

### Official Review · Reviewer_gNKs · 2025-11-01

**Soundness:** 3
**Presentation:** 3
**Contribution:** 2
**Rating:** 4
**Confidence:** 4

**Summary:**

The paper proposes an adversarial prompt tuning method termed AMPT, which utilizes multiple prompts with a conditional router mechanism during training. Its key findings are: (1) Using multiple prompts is more effective than using a single long prompt. (2) The conditional router outperforms a simple averaging approach. While this paper provides a reasonable approach for improving existing APT methods, its contributions are ultimately incremental, and the empirical gains do not appear to justify the significant increase in computational cost.

**Strengths:**

1.  It can be inspiring for improving the robustness of VLMs, especially regarding the use of multiple prompts.
2. The proposed method is easy to follow and intuitive.

**Weaknesses:**

1. The novelty is limited. The paper states that utilizing multiple prompts with a router mechanism, as opposed to a single long prompt, can lead to better robustness in adversarial prompt tuning. However, the proposed method feels like a straightforward application of a conditional weighted sum, which is a widely used technique.

2. The experimental setting needs to be clarified, and more experiments could be added to strengthen the authors' claims. Further details are provided in the "Questions" section.

3. The computational cost of this method is high relative to the gain in robustness. The improvements over the state-of-the-art APT baseline are modest: in the all-data setting (Table 1), AMPT achieves an average improvement of only 2.16% on PGD and 1.58% on AutoAttack (AA). In the 16-shot setting (Table 4), this gap shrinks even further to 0.68% on PGD and 0.87% on AA.

**Questions:**

1. Does this trend hold for other prompt lengths? The paper argues that using multiple prompts is better than a single long prompt by analyzing cases where the product (number of prompts × prompt length) is 32 and 64. Could you also provide results for other settings, such as when the product is 16 or 128?

2. What is the effect of the maximum perturbation budget on your findings? For example, with a maximum perturbation of 8/255, would the optimal number of prompts change? How would this convergence point (e.g., an optimal length of 8) differ from the one discussed in the paper?

3. Could you please clarify the prompt lengths used in Table 5?

---

### Note · Authors · 2025-11-12

I have read and agree with the venue's withdrawal policy on behalf of myself and my co-authors.